# Dengue virus-reactive CD8[+] T cells mediate cross-protection against subsequent Zika virus challenge

Jinsheng Wen[1,2], Annie Elong Ngono[1], Jose Angel Regla-Nava[1], Kenneth Kim[1], Matthew J. Gorman[3], Michael S. Diamond[3] & Sujan Shresta[1,2,4]

Zika virus (ZIKV) and dengue virus (DENV) are antigenically related flaviviruses that share cross-reactivity in antibody and T cell responses, and co-circulate in increasing numbers of countries. Whether pre-existing DENV immunity can cross-protect or enhance ZIKV infection during sequential infection of the same host is unknown. Here, we show that DENV-immune *Ifnar1*[−/−] or wild-type C57BL/6 mice infected with ZIKV have cross-reactive immunity to subsequent ZIKV infection and pathogenesis. Adoptive transfer and cell depletion studies demonstrate that DENV-immune CD8[+] T cells predominantly mediate cross-protective responses to ZIKV. In contrast, passive transfer studies suggest that DENV-immune serum does not protect against ZIKV infection. Thus, CD8[+] T cell immunity generated during primary DENV infection can confer protection against secondary ZIKV infection in mice. Further optimization of current DENV vaccines for T cell responses might confer cross-protection and prevent antibody-mediated enhancement of ZIKV infection.

---

[1] Division of Inflammation Biology, La Jolla Institute for Allergy & Immunology, La Jolla, CA 92037, USA. [2] Institute of Arboviruses, School of Basic Medical Sciences, Wenzhou Medical University, Wenzhou, Zhejiang 325000, China. [3] Department of Medicine, Molecular Microbiology, Pathology and Immunology, The Andrew M. and Jane M. Bursky Center for Human Immunology and Immunotherapy Programs, Washington University School of Medicine, St. Louis, MO 63110, USA. [4] Department of Medicine, School of Medicine, University of California, La Jolla, San Diego, CA 92037, USA. Correspondence and requests for materials should be addressed to S.S. (email: sujan@lji.org)

The closely related flaviviruses dengue virus serotypes 1–4 (DENV1–4) and Zika virus (ZIKV) share the *Aedes aegypti* vector and co-circulate in overlapping geographic ranges[1–3]. Cases of ZIKV infection in DENV-immune individuals, DENV infection of ZIKV-immune individuals, and even concurrent infection with both viruses are inevitable[4]. DENV and ZIKV share 52–57% total amino acid homology[5,6], and their immunologic cross-reactivity has been described previously[7–11]. Passive transfer experiments using plasma from DENV immune hosts have indicated that DENV antibodies (Abs) can enhance ZIKV pathogenesis[12]. However, little is known about DENV–ZIKV heterologous immune responses in the context of a sequentially infected host, and whether these responses may provide protection against or contribute to pathogenesis.

ZIKV infection during pregnancy has been shown to result in congenital malformations including microcephaly[13], whereas in adults infection is associated with encephalitis[14], genital tract infection and sexual transmission[14,15], Guillain Barré Syndrome[16], and immune-mediated thrombocytopenia[17]. DENV infection is associated with a range of clinical severity from asymptomatic illness to life-threatening dengue hemorrhagic fever/dengue shock syndrome[18]. Epidemiologic studies indicated that this severe form of DENV infection is most commonly associated with secondary heterotypic infection[19,20], in which an individual is infected by a second heterotypic DENV serotype following seroconversion to at least one other serotype. Mechanistically, the non-mutually exclusive hypotheses of antibody-dependent enhancement (ADE) and T cell original antigenic sin[21] have been proposed to explain why infection with a first virus can increase disease severity upon future infection with a second antigenically related virus.

Thorough epidemiological studies that characterize human DENV/ZIKV cross-reactive immune responses will take years to complete. However, laboratory evidence suggests that DENV and ZIKV cross-reactive Abs can reciprocally promote ADE of ZIKV[9,10,12,22] and DENV[8,23]. Consequently, vaccines for DENV, ZIKV, and other cross-reactive flaviviruses could sensitize individuals to more severe infection with a heterologous flavivirus[24–26]. Although vaccinology continues to focus on optimizing durable humoral immunity, evidence of ADE and T cell original antigenic sin in the contexts of sequential flavivirus infection or flavivirus immunogen exposure mandates a comprehensive interrogation of heterologous immunity and the crucial mechanisms responsible for protective vs. harmful immune responses.

Although initial studies supported a role for pathogenic, serotype cross-reactive T cells in promoting original antigenic sin in DENV infection[27–31], more recent data indicate a protective role for T cells is HLA-linked. CD8+ T cells are activated in DENV-infected patients[32,33], and DENV-immune individuals have both serotype-specific as well as cross-reactive CD8+ T cells that produce IFNγ and TNF, and exhibit cytotoxic functionality[27–29,34,35]. Additionally, recent studies have revealed that the magnitude and breadth of DENV-specific CD8+ T cell responses are associated with HLA alleles that correlate with clinical dengue disease[36,37].

Findings in mouse models have suggested a protective role for CD8+ T cells in DENV and ZIKV infection. A recent study in type I interferon (IFN) receptor (IFNAR)-deficient HLA-B*0702 and HLA-A*0101 transgenic mice demonstrated that CD8+ T cells primed with cross-reactive DENV peptide epitopes could have protective activity against ZIKV[38]. Another study in *LysM-Cre+Ifnar1fl/fl* C57BL/6 mice, which lack IFNAR in a subset of myeloid cells and possesses IFNAR-competent T cells, showed that depletion of CD8+ T cells results in increased ZIKV replication, ZIKV-specific CD8+ T cells have cytotoxic activity in vivo, and

adoptive transfer of ZIKV-primed CD8+ T cells reduces ZIKV replication[39]. Prior studies using models of DENV infection in C57BL/6 and 129/Sv mice globally lacking IFNAR or both type I and II IFN receptors have used similar loss-of-function (CD8+ T cell depletion) and gain-of-function (CD8+ T cell transfer and peptide immunization) approaches to demonstrate a critical role for CD8+ T cells in protection against DENV infection and disease[40–42]. Additionally in the context of secondary DENV infections, studies in these IFNAR-deficient mice have revealed that CD8+ T cells are required for protection against heterotypic, but not homotypic, secondary DENV infection[43] and that CD8+ T cells can confer protection against DENV infection even under ADE conditions[44]. Collectively, these results support roles for CD8+ T cells in cross-protection against DENV and ZIKV infection. Notwithstanding these studies, the following key questions have not been answered: Does previous DENV exposure confer cross-protection against ZIKV, as observed in the context of heterotypic reinfection with different DENV serotypes? What are the roles of cellular vs. humoral immunity in mediating such cross-protection against ZIKV?

Here, we explored the clinical and virological outcomes, and explored the immunological mechanisms in DENV-immune mice subsequently challenged with ZIKV. We show that DENV immunity can confer protection against ZIKV infection in the same host. By depleting naive CD8+ T cells and transferring DENV-immune serum or CD8+ T cells, we demonstrated that CD8+ T cells, and not DENV-reactive Abs in serum, mediate cross-protection against ZIKV infection. These findings have implications for DENV and ZIKV vaccine development efforts.

## Results

**Cross-reactivity of ZIKV CD8+ T cell epitopes to DENV2.** We recently identified ZIKV-derived peptide epitopes recognized by CD8+ T cells in H-2b mice[39]. We used this information to identify ZIKV epitopes that were cross-reactive with DENV in congenic H-2b *Ifnar1−/−* mice. *Ifnar1−/−* mice were used because ZIKV does not efficiently inhibit type I IFN receptor signaling or replicate enough to cause disease pathogenesis in adult wild-type (WT) H-2b mice[45]. *Ifnar1−/−* mice were infected with DENV2 strain S221, followed by detection of peptide-specific IFNγ-secreting CD3+ CD8+ T cells in the spleen via intracellular cytokine staining (ICS) assay on day 7 after infection. The percentages of CD3+CD8+IFNγ+ T cells in the spleen from DENV-infected mice after stimulation with ZIKV peptides from prM$_{44-52}$, E$_{4-12}$, E$_{7-15}$, NS3$_{347-355}$, and NS5$_{18-27}$ were $0.67 \pm 0.15\%$, $0.55 \pm 0.1\%$, $0.59 \pm 0.14\%$, $0.58 \pm 0.14\%$, and $0.59 \pm 0.12\%$, respectively (mean $\pm$ SEM, two independent experiments, Supplementary Fig. 1a). CD3+CD8+ T cells directed to ZIKV epitopes prM$_{44-52}$, E$_{7-15}$, NS3$_{347-355}$, and NS5$_{18-27}$ also co-expressed IFNγ and TNF (Supplementary Fig. 1b) and IFNγ and CD107a (Supplementary Fig. 1c), exhibiting a polyfunctional phenotype. Altogether, these results identify a total of six H-2b-restricted, CD8+ T cell epitopes in ZIKV that are cross-reactive with DENV.

**Cross-reactivity of DENV2 CD8+ T cell epitopes to ZIKV.** The percent identity between the six ZIKV-derived epitopes that are cross-reactive with DENV2 and corresponding DENV2 peptides ranged from 44 to 89% (Table 1). To assess the reciprocal cross-reactivity of DENV2 epitopes to ZIKV, *Ifnar1−/−* mice were inoculated with ZIKV strain FSS13025 and splenocytes were harvested on day 7 post infection to measure antigen-specific T cell responses via ICS assay upon stimulation with five different DENV2 peptide variants (DENV2-prM$_{44-52}$, DENV2-E$_{4-12}$, DENV2-E$_{7-15}$, DENV2-NS3$_{347-355}$, and DENV2-NS5$_{18-27}$)

**Table 1 ZIKV epitopes and DENV2 variants**

| Peptides[a] | Sequences[b] | Conservation[c] |
|---|---|---|
| **MR-prM$_{20-28}$** | **ISFATTLGV** | 44% |
| DENV2-prM$_{20-28}$ | LLFKTEDGV | |
| **FSS/MR-prM$_{44-52}$** | **ATMSYECPM** | 44% |
| DENV2-prM$_{44-52}$ | DTITYKCPL | |
| **FSS/MR-E$_{4-12}$** | **IGVSNRDFV** | 89% |
| DENV2-E$_{4-12}$ | IGISNRDFV | |
| **FSS/MR-E$_{7-15}$** | **SNRDFVEGM** | 89% |
| DENV2-E$_{7-15}$ | SNRDFVEGV | |
| **FSS/MR-NS3$_{347-355}$** | **PSVRNGNEI** | 56% |
| DENV2-NS3$_{347-355}$ | PSIKAGNDI | |
| **FSS/MR-NS5$_{18-27}$** | **CAEAPNMKII** | 50% |
| DENV2-NS5$_{18-27}$ | ESEVPNLDII | |

[a] ZIKV peptides in bold are positive as determined via ICS assays in DENV2-infected mice
[b] Amino acid residues underlined are conserved between the ZIKV epitope and DENV2 variant
[c] % shared amino acids between ZIKV and DENV2

(Table 1). The frequencies of CD3$^+$CD8$^+$IFNγ$^+$ T cells after ZIKV infection that were restimulated ex vivo with DENV2-E$_{4-12}$, DENV2-E$_{7-15}$, and DENV2-NS3$_{347-355}$ were 2.92 ± 0.18%, 0.56 ± 0.1%, and 0.55 ± 0.11%, respectively (two independent experiments, Supplementary Fig. 2a). CD3$^+$CD8$^+$ T cells stimulated with the DENV2-E$_{4-12}$ peptide also co-expressed IFNγ and TNF (Supplementary Fig. 2b) and IFNγ and CD107a (Supplementary Fig. 2c). Thus, of five H-2$^b$-restricted, DENV2-derived CD8$^+$ T cell epitopes tested, four exhibited cross-reactivity with ZIKV.

**CD8$^+$ T cell-dependent DENV2 immunity reduces ZIKV burden**. To explore the role of DENV immunity in protection vs. pathogenesis during ZIKV infection, we challenged DENV2-immune Ifnar1$^{-/-}$ mice (28 days post infection) with ZIKV strain FSS13025 and determined antigen-specific CD8$^+$ T cell responses and viral titers in sera and tissues at day 3 after ZIKV challenge. Day 3 post ZIKV challenge was chosen for analysis of the T cell and viral phenotype to focus our study on the adaptive memory but not primary T cell response; this decision was based on prior studies demonstrating that the CD8$^+$ T cell response to primary DENV or ZIKV infection in mice is not observed on day 3 but becomes detectable after day 5 of infection[39–41]. Accordingly, a cross-reactive peptide-specific CD8$^+$ T cell immune response was absent in naive (Fig. 1a, c) but present in DENV2-immune mice (Fig. 1b). Infectious ZIKV levels in the serum, liver, brain, eye, and testis of DENV2-immune mice were 25-fold, 91-fold, 273-fold, 25-fold, and 383-fold lower, respectively, than in naive mice (isotype DENV2-immune vs. isotype Naive in Fig. 1e–i). When a depleting anti-mouse CD8 monoclonal Ab was administered, the majority of cross-reactive peptide-specific CD8$^+$ T cells were absent (Fig. 1d), and infectious ZIKV levels in the serum, liver, brain, eye, and testis of CD8$^+$ T cell-depleted DENV2-immune mice were 159-fold, 174-fold, 424-fold, 17-fold, and 217-fold higher than isotype control Ab-treated, CD8$^+$ T cell-sufficient DENV2-immune mice (anti-CD8 DENV2-immune vs. isotype DENV2-immune in Fig. 1e–i). Notably, infectious ZIKV in the serum from CD8$^+$ T cell-depleted DENV2-immune mice was not the same as in naive mice (anti-CD8 DENV2-immune vs. isotype Naive and anti-CD8 Naive in Fig. 1e); rather, ZIKV levels in sera of CD8$^+$ T cell-depleted DENV2-immune mice was higher (5–6-fold, two-tailed Mann–Whitney test, $P < 0.01$) than in both groups of naive mice. Taken together, these results demonstrate that prior DENV immunity confers cross-protection against ZIKV infection through a CD8$^+$ T cell-dependent mechanism. Based on a published study demonstrating that CD8$^+$ T cells can prevent DENV ADE[44], the higher level of ZIKV in the serum of

CD8$^+$ T cell-depleted DENV2-immune mice relative to naive mice suggests that CD8$^+$ T cells also might control the ADE induced by cross-reactive DENV Abs following ZIKV infection.

**DENV2 immune sera-mediated neutralization of ZIKV in vitro**. To evaluate the role of the DENV-immune Ab response in mediating protection or enhancement during ZIKV infection, we collected sera from naive and DENV2-infected Ifnar1$^{-/-}$ mice on day 28 after infection. We first tested the DENV2-immune sera for their ability to bind DENV2 and ZIKV by ELISA. DENV2-immune sera bound to both DENV2 and ZIKV, with lower endpoint titers (EPTs) against ZIKV than DENV2 (Supplementary Fig. 3). We next assessed whether the DENV2-immune sera could neutralize DENV2 and ZIKV in vitro using a flow cytometry-based neutralization assay[46]. As expected, naive mouse serum failed to neutralize DENV2 or ZIKV (Fig. 2a). In comparison, DENV2-immune serum efficiently neutralized DENV2, with 50% neutralization titer (NT$_{50}$) of ~1:180. DENV2-immune serum could also neutralize ZIKV weakly, with an NT$_{50}$ of ~1:15. These results demonstrate that DENV2-immune sera can bind and neutralize both DENV2 and ZIKV in vitro, although levels are lower against ZIKV relative to DENV2.

**DENV2-immune sera do not limit ZIKV infection in vivo**. We next examined the capacity of DENV2-immune sera to neutralize ZIKV in Ifnar1$^{-/-}$ and WT mice. High volumes (300 or 600 μl) of naive mouse serum, and low (33 μl), medium (100 μl), and high volumes (300 or 600 μl) of DENV2-immune serum were transferred passively via retro-orbital route to naive Ifnar1$^{-/-}$ mice. One day after the serum transfer, mice were challenged with ZIKV, followed by analysis of the splenic CD8$^+$ T cell response and ZIKV replication in various tissues on day 3 after the viral challenge. As expected, at this early time point after ZIKV challenge in naive settings, no significant peptide-specific CD8$^+$ T cell responses were detected in all mouse groups (Supplementary Fig. 4). Passive transfer of DENV2-immune serum from Ifnar1$^{-/-}$ donor mice did not reduce infectious ZIKV levels in any tissues of Ifnar1$^{-/-}$ recipient mice, and if anything, slight enhancement was observed in the serum of mice transferred with 300 μl of immune serum (Fig. 2b). When we increased the serum volume to 600 μl, we observed neither neutralization nor enhancement (Fig. 2c). In parallel experiments, high volumes (75 μl) of naive WT mouse serum, and high (75 μl), medium (25 μl), and low volumes (8.3 μl) of DENV2-immune serum obtained from WT mice were transferred passively via retro-orbital route to 2-week-old WT mice (Supplementary Fig. 5a, b). In a separate study with 4-week-old WT recipients, 200 μl of naive or DENV2-immune serum obtained from WT mice was retro-orbitally transferred (Supplementary Fig. 5c, d). Similar to results obtained from experiments with DENV2-immune serum obtained from Ifnar1$^{-/-}$ mice, DENV2-immune serum harvested from WT mice did not decrease ZIKV levels in tissues. These results suggest that DENV-immune Abs play a minimal role in cross-protection against subsequent ZIKV infection.

**DENV2-reactive CD8$^+$ T cells protect against ZIKV challenge**. Given the absence of protection after serum transfer, we assessed whether DENV-exposed CD8$^+$ T cells were sufficient to control ZIKV infection. Ifnar1$^{-/-}$ mice were inoculated with DENV2 for 28 days, followed by isolation of DENV2-exposed CD8$^+$ memory T cells from the spleen. $1 \times 10^7$ DENV2-exposed splenic CD8$^+$ T cells then were transferred adoptively to naive Ifnar1$^{-/-}$ recipient mice. High numbers of cross-reactive peptide-specific CD8$^+$ T cells were detected in mice receiving $1 \times 10^7$ DENV2-exposed CD8$^+$ T cells, as compared to mice receiving $1 \times 10^7$ naive CD8$^+$

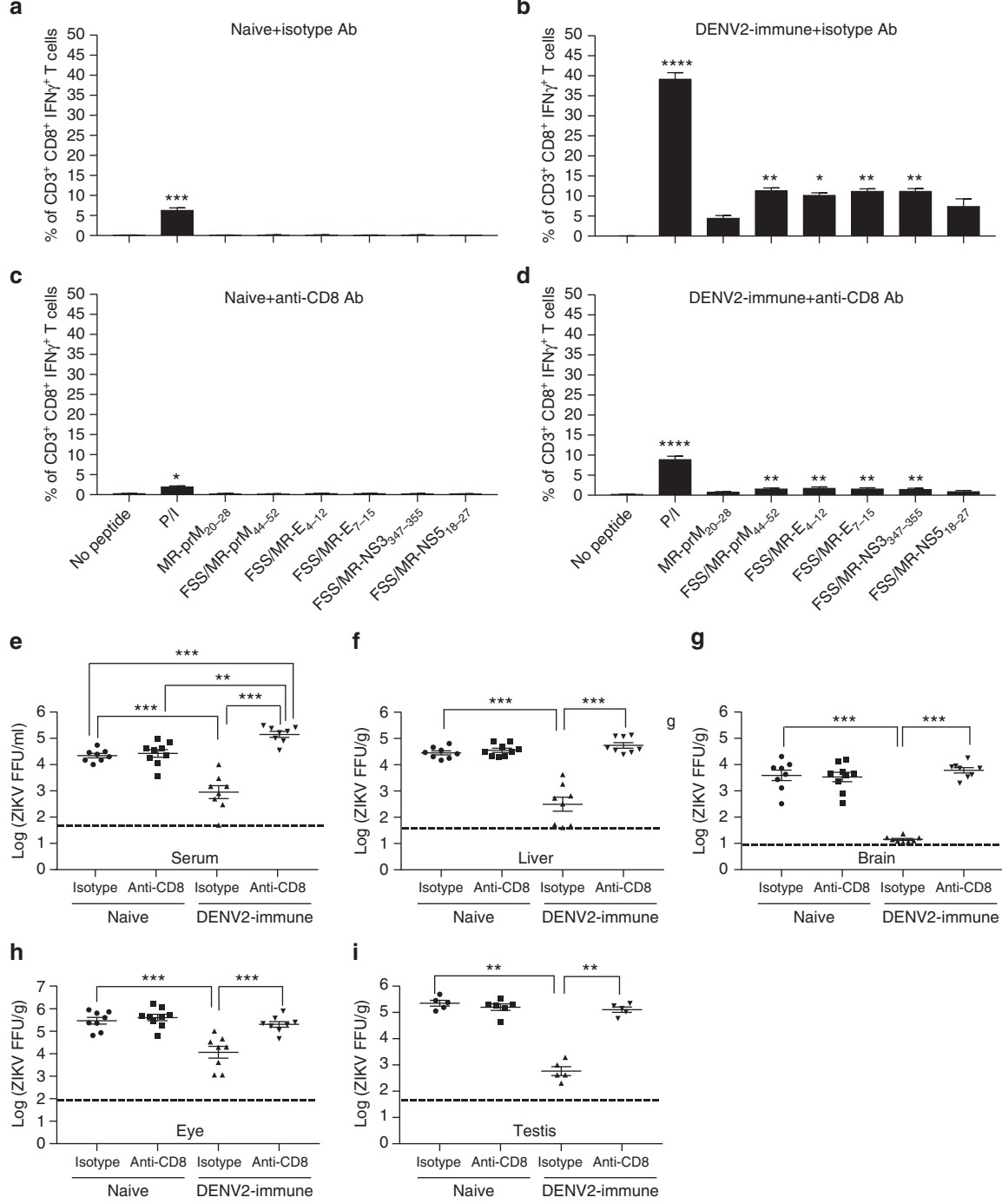

**Fig. 1** ZIKV tissue burden in DENV-immune *Ifnar1*[−/−] mice in the presence or absence of CD8[+] T cells. *Ifnar1*[−/−] mice were immunized intraperitoneally with DENV2 ($2 \times 10^2$ FFU) for 28 days. Naive and DENV2-immune mice were injected intraperitoneally with isotype control Ab and anti-mouse CD8 Ab at 3 days and 1 day before ZIKV challenge. Three days post ZIKV challenge, peptide-specific CD3[+]CD8[+]IFNγ[+] T cells (**a–d**; **a**, **c** naive mice; **b**, **d** DENV2-immune mice) and infectious ZIKV levels in sera (**e**) and indicated organs (**f–i**) were detected using ICS assay and FFA, respectively. Data were pooled from two independent experiments with $n = 3$–5 mice per group per experiment and expressed as mean ± SEM. *$p < 0.05$, **$p < 0.01$, ***$p < 0.001$, ****$p < 0.0001$. A Kruskal–Wallis one-way ANOVA was used for **a–d** while two-tailed Mann–Whitney test was used for **e–i**. Supplementary Tables 1 and 2 provide exact values of *n* and *p*

T cells (Fig. 3a). In serum and all organs, transfer of $1 \times 10^7$ DENV2-exposed CD8[+] T cells led to significant reduction in infectious ZIKV levels (Fig. 3b). Consistent with these results with *Ifnar1*[−/−] mice, adoptive transfer of DENV-exposed CD8[+] T cells isolated from WT mice also resulted in a significant decrease in ZIKV levels in the serum and liver of WT recipients (Fig. 4a, b). Thus, DENV-reactive CD8[+] T cells can mediate cross-protection

against ZIKV infection by reducing virus levels in multiple tissues.

We next evaluated whether DENV-exposed CD8[+] T cells were sufficient to protect against lethal ZIKV disease in naive *Ifnar1*[−/−] mice, which support high levels of ZIKV replication and succumb even at a low viral challenge dose[45]. Naive *Ifnar1*[−/−] mice received either $1 \times 10^7$ naive CD8[+] T cells or $1 \times 10^7$ DENV2-exposed

CD8[+] T cells from the spleen, followed by challenge of the recipient mice with $1 \times 10^3$ focus-forming unit (FFU) of ZIKV. Mice were observed for 2 weeks, and clinical score, weight loss, and death (or moribund condition mandating euthanasia) were recorded on a daily basis[15]. Mice receiving DENV2- exposed CD8[+] T cells exhibited less disease based on clinical scores (Fig. 5b), weight loss (Fig. 5c), and mortality (Fig. 5d) relative to mice receiving naive CD8[+] T cells (Fig. 5a–d). Indeed, 90% of mice with DENV2-exposed CD8[+] T cells survived compared to only 10% of mice that received naive CD8[+] T cells.

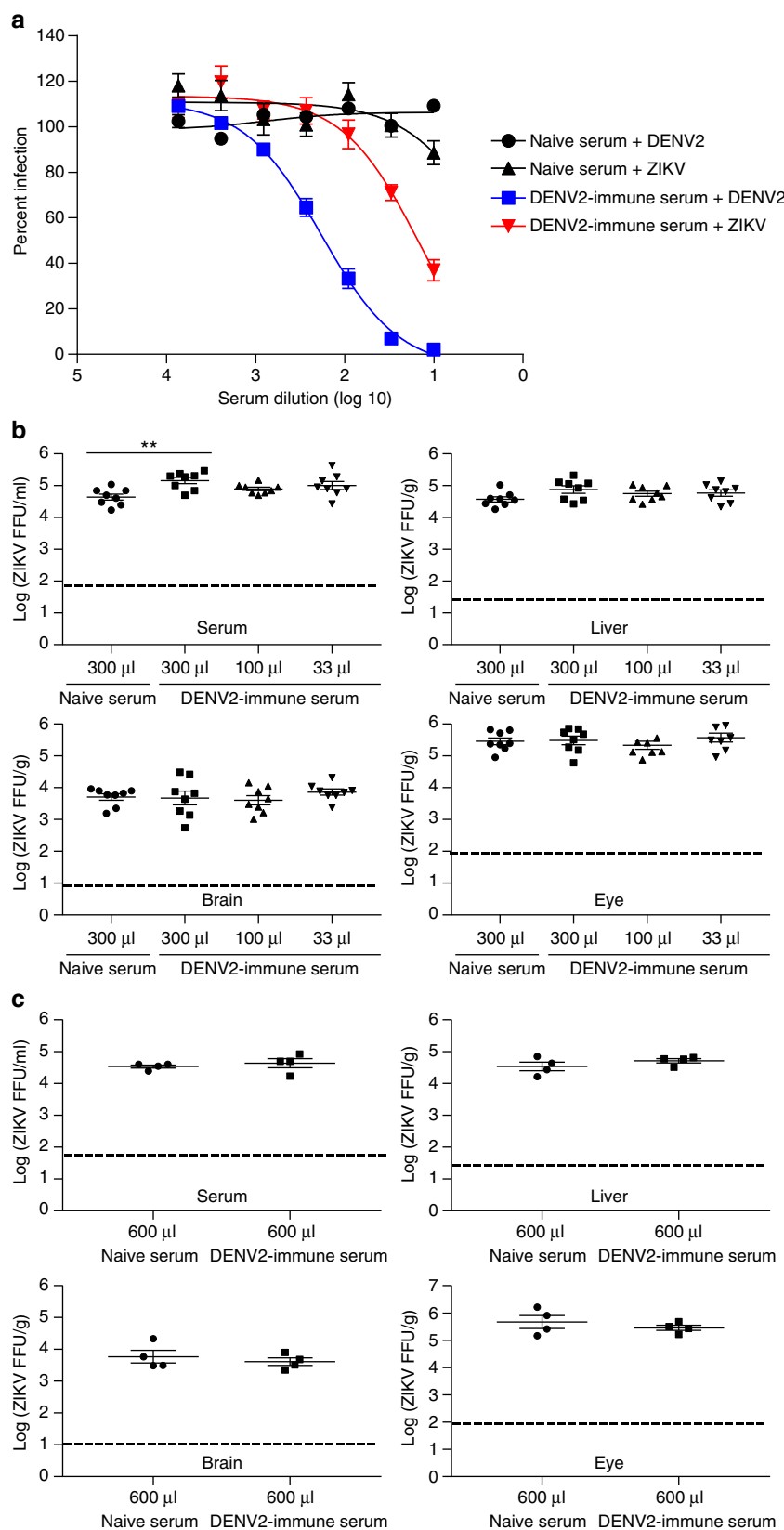

These results demonstrate that DENV-reactive CD8[+] T cells can confer cross-protection against lethal ZIKV infection in mice.

## Discussion

The close relationships between DENV and ZIKV at the amino acid sequence and serologic levels indicate possibilities for cross-protection or immune enhancement during sequential DENV and ZIKV infections. A study focusing on cross-reactive T cell responses in *Ifnar1*[−/−] HLA-transgenic mice has shown that prior DENV immunity alters immunodominance patterns of CD8[+] T cell responses to subsequent ZIKV infection[38], while another has demonstrated that passive transfer of DENV-immune plasma into naive *Stat2*[−/−] mice can enhance ZIKV infection and disease severity[12]. Based on these findings, the following question needs to be answered urgently: What is the role of prior DENV immunity during sequential ZIKV infection of the same host? Epidemiologic studies in humans will take years to perform, and the precise contributions of cellular vs. humoral immunity cannot be easily dissected in humans. The goals of our study were to determine the impact of DENV immunity on the outcome of subsequent ZIKV infection, and to identify the immune components responsible for any such influence. We observed that DENV immunity significantly reduced infectious ZIKV burden in sera and tissues. Humoral vs. CD8[+] T cell contributions were isolated and quantified via CD8[+] T cell depletion of DENV-immune mice, and adoptive transfer of DENV-immune serum or CD8[+] T cells to naive mice. DENV-immune mice contained high levels of DENV-binding and DENV-neutralizing Abs but low levels of ZIKV-binding and ZIKV-neutralizing Abs, as measured by ELISA and in vitro neutralization assays, and passive transfer experiments revealed no role for DENV-immune sera in protection against ZIKV in mice. In contrast, T cell depletion and adoptive transfer studies demonstrated that ZIKV protection was mediated principally by DENV-exposed CD8[+] T cells. This result is consistent with prior murine studies demonstrating that DENV-specific CD8[+] T cells can confer protection in the context of heterotypic DENV serotype and ADE infections[43,44]. A short period of cross-protection against heterotypic infection by other DENV serotypes has been well documented in humans[47–50]. This cross-protection in humans has been assumed to rely on high titers of cross-reactive Abs reactive for all DENV serotypes despite the lack of experimental evidence[51–53]. More recent studies examining the anti-DENV T cell response in humans support a protective role for T cells against DENV infection in humans[36,37,54], in agreement with published studies in mice[40–44,55,56]. Based on our results, we speculate that the anti-DENV CD8[+] T cell response in humans will mediate at least transient cross-protection against subsequent ZIKV infection[57].

At present, it remains unknown whether human DENV immunity is protective or pathogenic against ZIKV infection. Documentation of cross-reactivity between DENV and ZIKV in humoral[8–11] and cellular immune compartments[38] is growing. Although many studies have explored the Ab cross-reactivity between DENV and ZIKV, the exact role of prior DENV humoral immunity during subsequent ZIKV infection also is uncertain[58].

In vitro, cross-reactive murine DENV Abs can enhance ZIKV infection[22,59] and human DENV Abs can neutralize[5,9–11,60,61] or enhance[9,10,12,22,62,63] ZIKV infection depending on their epitope specificity and avidity. In vivo, human DENV Abs can protect mice against ZIKV infection[11,61] or mediate ADE[12]. These results are not surprising, as the anti-DENV Ab response can exhibit protection, enhancement, or no effect depending on epitope specificity, virion binding activity, concentration, and avidity, albeit the precise characteristics of the anti-flavivirus Ab response that produces these different outcomes are not fully understood. Neutralization vs. enhancement vs. no effect outcomes are functions of the stoichiometry of Ab binding to the viral particles and infection of cells expressing Fc-gamma receptors[64,65]. Results in our study demonstrate that DENV-immune Abs derived from mouse serum can neutralize ZIKV in vitro, but are not sufficient to protect against or enhance infection at the amounts we tested in vivo. The lack of ADE that we observed with DENV-immune mouse sera contrasts with enhanced ZIKV pathogenesis described in *Stat2*[−/−] mice that were passively transferred DENV-immune human plasma;[12] this disparity may be due to differences in the plasma-derived Ab preparations and/or mouse models (WT and *Ifnar1*[−/−] vs. *Stat2*[−/−]). Similar to our results, two recent rhesus macaque studies showed that DENV-immune serum could neutralize ZIKV infection in vitro and did not promote ADE against ZIKV in vivo[66,67]. One of these studies with rhesus macaques showed a reduction in the duration of ZIKV viremia in DENV-immune animals relative to naive controls, suggesting cross-protection[67]. However, in the second rhesus macaque study, prior DENV immunity failed to confer protection against subsequent ZIKV infection[66]. This lack of robust cross-protection in macaque may have several explanations: (a) DENV does not establish significant levels of infection in non-human primates[68] and thus may not induce a robust anti-DENV immune response; accordingly, the anti-DENV T cell response may not be protective, and the cross-reactive Ab response may be too weak to protect against or enhance ZIKV infection; (b) animals in both studies were challenged with ZIKV at 1–2 years after DENV infection. Thus, a transient period of cross-immunity may have waned by the time of ZIKV challenge. Studies using mice that have been exposed to DENV for greater lengths of time than 1 month (such as 3, 6, and 12 months) may help answer questions related to the window of cross-protection.

Our results agree with emerging literature implicating a complex interplay between humoral and cellular immunity in flavivirus-experienced individuals. There is evidence that some individuals mount weaker CD8[+] T cell responses than others; these weaker T cell responses, in combination with suboptimal Ab responses, may lead to severe disease manifestations, whereas an efficient T cell or Ab response alone may be sufficient for protection. Recent studies have revealed that the magnitude and breadth of DENV-specific CD8[+] T cell and CD4[+] T cell responses correlate with specific HLA alleles[36,37,54,69,70]. Investigations in HLA transgenic mice also showed that CD8[+] T cell responses to ZIKV are stronger in HLA-B*0702 mice than HLA-A*0101 mice[38]. However, the net impact of HLA and T cell responses on ZIKV infection and pathogenesis can only be determined through

**Fig. 2** The effect of DENV2-immune sera on ZIKV infection in vitro and in *Ifnar1*[−/−] mice. *Ifnar1*[−/−] mice were immunized intraperitoneally with DENV2 ($2 \times 10^2$ FFU) for 28 days. Naive *Ifnar1*[−/−] mouse sera ($n = 10$) and DENV2-immune *Ifnar1*[−/−] mouse sera ($n = 10$) were examined for their ability to neutralize DENV2 or ZIKV using U937 DC-SIGN cells and a flow cytometry-based neutralization assay (**a**). Data were pooled from two independent experiments with $n = 5$ mouse sera per group and expressed as mean ± SEM. Different volumes of naive and DENV2-immune *Ifnar1*[−/−] mouse sera were passively transferred (retro-orbital route) to naive recipient *Ifnar1*[−/−] mice (**b**, **c**). Three days post retro-orbital challenge with $1 \times 10^4$ FFU of ZIKV FSS13025, viral burdens in sera and indicated organs were measured using FFA. In **b**, data are pooled from two independent experiments ($n = 3–4$ mice per group per experiment), while results in **c** represent one experiment. Data are expressed as mean ± SEM. **p < 0.01. **b** A Kruskal–Wallis one-way ANOVA. **c** Two-tailed Mann–Whitney test. Supplementary Tables 1 and 2 provide exact values of $n$ and $p$

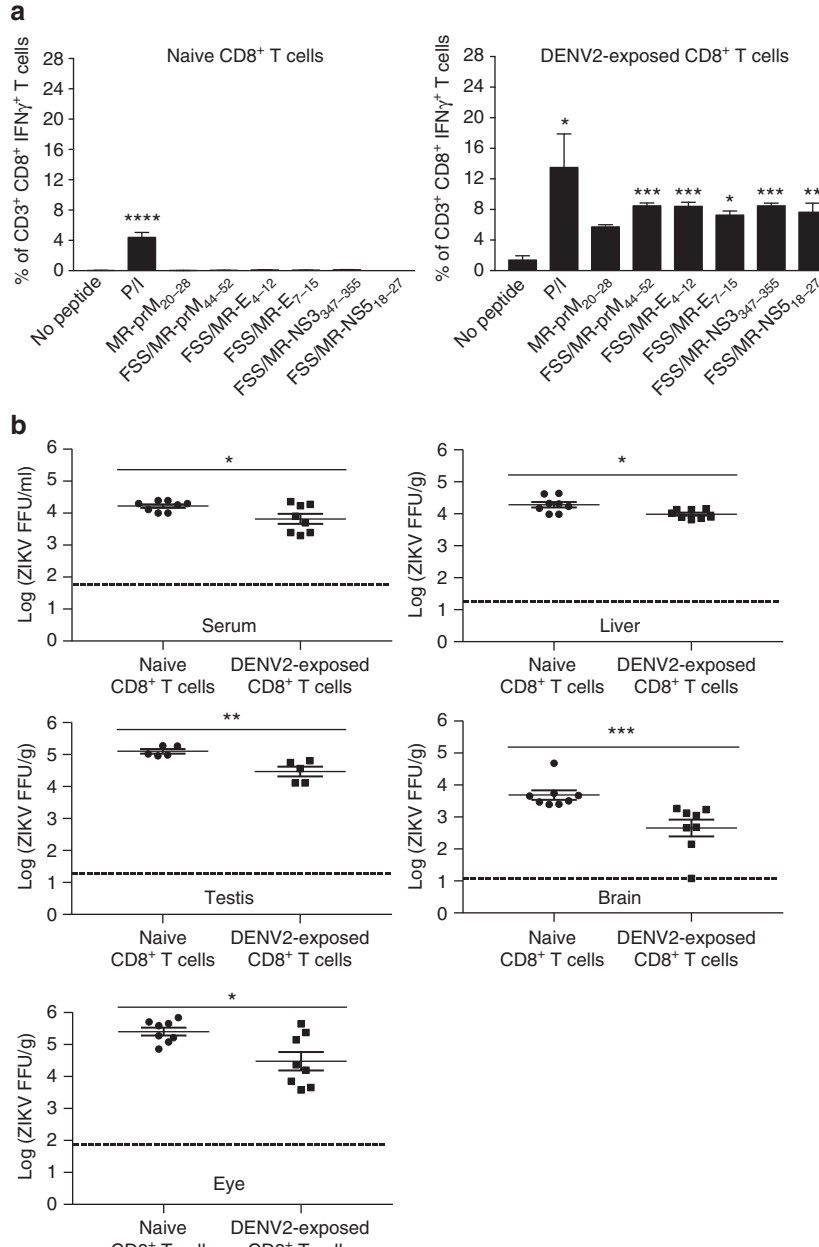

**Fig. 3** Reduced ZIKV burden in *Ifnar1*$^{-/-}$ mice adoptively transferred with DENV2-exposed CD8$^+$ T cells. Naive (1 × 10$^7$) and DENV2-exposed CD8$^+$ T cells (1 × 10$^7$) from *Ifnar1*$^{-/-}$ mice were adoptively transferred to naive recipient *Ifnar1*$^{-/-}$ mice. Three days post ZIKV challenge (1 × 10$^4$ FFU of ZIKV FSS13025, retro-orbital route), cross-reactive peptide-specific CD8$^+$ T cell responses (**a**) and ZIKV levels in sera and indicated organs (**b**) were detected by ICS assay and FFA, respectively. Data are pooled from two independent experiments (*n* = 3-4 mice per group per experiment) and expressed as mean ± SEM. *$p < 0.05$, **$p < 0.01$, ***$p < 0.001$,****$p < 0.0001$. **a** A Kruskal–Wallis one-way ANOVA. **b** Two-tailed Mann–Whitney test. Supplementary Tables 1 and 2 provide exact values of *n* and *p*

a combination of clinical studies, epidemiologic studies, and vaccine and antiviral trials.

If some individuals have genetic or acquired predispositions to weaker CD8$^+$ T cell responses to viruses such as DENV and ZIKV, these people may not be protected in the scenarios of sequential flavivirus infection or vaccination followed by heterologous infection. An inadequate CD8$^+$ T cell response in the presence of enhancing levels of Abs may account for some of the worsened dengue outcomes seen in individuals vaccinated for Japanese encephalitis virus[25] or a heterologous DENV serotype[26]. Based on findings of the present study and two recent publications[12,38], an inefficient CD8$^+$ T cell response,

in combination with ADE, might have contributed to severe ZIKV disease manifestations in individuals with prior DENV exposure during the 2015–2016 ZIKV epidemic in Latin America. As the geographic ranges of flaviviruses continue to expand and merge, and human hosts travel globally, flavivirus vaccines that are engineered to induce both robust CD8$^+$ T cell and Ab responses may be critical to achieve maximal efficacy and safety.

## Methods

**Mice and ethics statement**. *Ifnar1*$^{-/-}$ (C57BL/6 background) and WT C57BL/6 mice were originally obtained from Dr. W. Yokoyama (Washington University in

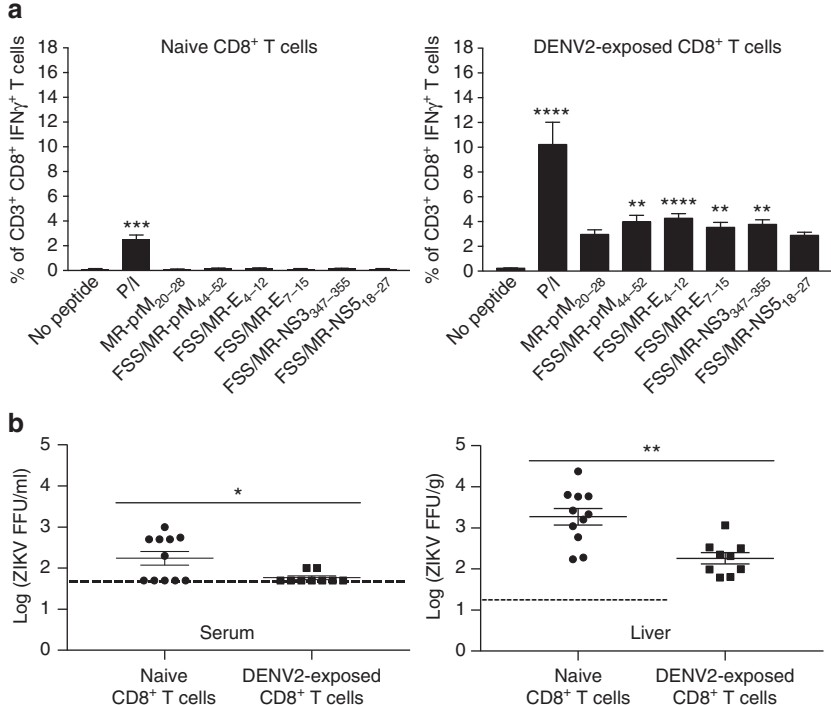

**Fig. 4** Reduced ZIKV burden in WT mice adoptively transferred with DENV2-exposed CD8+ T cells. Naive or DENV2-exposed CD8+ T cells ($1 \times 10^7$) from WT mice were adoptively transferred to naive recipient WT mice that were intraperitoneally injected with 2 mg of the IFNAR1-blocking mAb MAR1-5A3 1 day before ZIKV challenge. Three days post ZIKV challenge ($1 \times 10^6$ FFU of FSS13025, retro-orbital route), cross-reactive peptide-specific CD8+ T cell responses (**a**) and ZIKV levels in serum and liver (**b**) were detected by ICS assay and FFA, respectively. Data are pooled from two independent experiments ($n = 4$–6 mice per group per experiment) and expressed as mean ± SEM. *$p < 0.05$, **$p < 0.01$, ***$p < 0.001$, ****$p < 0.0001$. **a** A Kruskal–Wallis one-way ANOVA. **b** Two-tailed Mann–Whitney test. Supplementary Tables 1 and 2 give exact values of $n$ and $p$

St. Louis) via Dr. Carl Ware (La Jolla Institute for Allergy and Immunology) and the Jackson Laboratory (USA), respectively. All mice were maintained and bred at the La Jolla Institute for Allergy & Immunology under standard pathogen free conditions. All mouse experiments were approved by the Institutional Animal Care and Use Committee under protocol #AP028-SS1-0615. Sample sizes were estimated based on previous similar studies. Animal experiments were not randomized or blinded.

**Peptide synthesis.** H-2b-restricted CD8+ T cell epitopes in ZIKV were identified previously[39] (Supplementary Table 3). Five DENV2 variants of the ZIKV epitopes that stimulated cross-reactive DENV2-exposed CD8+ T cells also were selected. Peptides were synthesized by Synthetic Biomolecules with a purity >95% confirmed by mass spectrometry. All peptides were dissolved in DMSO to 40 mg/ml and stored at −80 °C.

**Viruses, cells, and reagents.** ZIKV Asian lineage strain FSS13025 (Cambodia, 2010) was obtained from the World Reference Center for Emerging Viruses and Arboviruses (WRCEVA) and propagated in C6/36 *Aedes albopictus* cells (ATCC® CRL-1660™). The mouse-adapted DENV2 strain S221 is derived from DENV2 D2S10[40]. Baby Hamster Kidney (BHK)-21 cells and U937 DC-SIGN cells were purchased from the American Type Culture Collection (ATCC). ZIKV and DENV2 titers were analyzed on BHK-21 cells using focus-forming assay (FFA) and expressed as FFU/ml. PE-conjugated mouse anti-human CD209 monoclonal antibody (mAb) (clone DCN46) and Brefeldin A (BFA) were purchased from BD Biosciences. DENV-specific mAb 2H2 (clone D3-2H2-9-21) and pan Flavivirus-specific mAb 4G2 hybridoma (clone D1-4G2-4-15) were purchased from the ATCC, and purified mAbs were obtained from BioXCell. Ab used for mouse CD8+ T cell depletion (rat anti-mouse CD8 mAb, clone 2.43) and isotype control Ab (rat IgG2, clone LTF-2) were purchased from BioXCell. IFNAR1-blocking mAb MAR1-5A3 also was purchased from BioXCell. Mouse CD8+ T cell magnetic positive selection microbeads were purchased from Miltenyi Biotec. Rat anti-mouse CD3 PerCpCy 5.5 (clone 145-2C11) and anti-IFNγ FITC (clone XMG 1.2) were purchased from Tonbo Biosciences. Rat anti-mouse CD8 PE-Cy7 (clone 53–67), anti-CD107a PE (clone 1D4B), anti-TNF APC (clone MP6-XT22), 3,3′,5,5′-Tet-ramethylbenzidine (TMB), and Phorbol 12-myristate 13-acetate (PMA)/Ionomycin were purchased from eBioscience. Horseradish peroxidase (HPR)-conjugated goat anti-mouse IgG and carboxymethyl cellulose (CMC) were purchased from Sigma. True Blue substrate was purchased from KPL. The ZIKV envelope (E) protein was purchased from the Native Antigen Company.

**Mouse infections.** Five-to-six-week-old *Ifnar1*−/− male and female mice were infected with $2 \times 10^4$ FFU of DENV2 or $1 \times 10^2$ FFU of ZIKV via retro-orbital route. After 7 days, mice were sacrificed, and splenocytes were used for an ICS assay. In addition, 5-to-6-week-old *Ifnar1*−/− mice were inoculated with $2 \times 10^2$ FFU DENV2 via intra-peritoneal route for 28 days, and these DENV2-immune mice were used for CD8+ T cell depletion, CD8+ T cell isolation, or serum collection. Five-to-six-week-old WT mice were administered 2 mg IFNAR1-blocking mAb MAR1-5A3 intraperitoneally 1 day before inoculation with $1 \times 10^4$ FFU of DENV2 via retro-orbital route. After 28 days, these DENV2-immune WT mice were used for CD8+ T cell isolation or serum collection.

**ZIKV challenge of CD8+ T cell-depleted DENV2-immune mice.** Age-matched naive and DENV2-immune *Ifnar1*−/− mice were injected intraperitoneally with either rat anti-mouse CD8 Ab (250 μg/mouse) or isotype control Ab (250 μg/mouse, rat IgG2) at day-3 and day-1 before ZIKV challenge. *Ifnar1*−/− mice were challenged retro-orbitally with $1 \times 10^4$ FFU of ZIKV FSS13025. Three days after ZIKV challenge, mice were sacrificed, and serum and spleen were harvested. Spleens were used for ICS assay. After cardiac perfusion with PBS, liver, testis, brain, and eye were harvested. ZIKV titers in sera and other tissues of *Ifnar1*−/− mice were quantified using a FFA.

**DENV2-binding and ZIKV-binding IgG ELISA.** Sera were harvested from naive or DENV2-immune *Ifnar1*−/− mice on day 28 after infection. Ninety six-well ELISA plates were incubated with either UV-inactivated DENV2 ($2 \times 10^4$ FFU/well) or ZIKV E protein (1 μg/ml) in 50 μl coating buffer 0.1 M NaHCO₃ overnight at 4 °C. The plates were then washed to remove unbound virus or ZIKV protein and blocked for 1 h at room temperature (RT). Sera were diluted to 1:30, 1:90, 1:270, 1:810, 1:2430, 1:7290, 1:21870, and 1:65610, and added to individual wells (100 μl/well). After 1.5 h of incubation at RT, plates were washed and bound Ab was detected using HRP-conjugated goat anti-mouse IgG (at a dilution of 1:500). The plates were developed with 3,3′,5,5′-TMB and the absorbance was read at 450 nm. EPTs were defined as reciprocal serum dilutions that corresponded to two times the average OD values obtained with naive serum.

**Serum neutralization assay.** Flow cytometry-based assay was used to evaluate neutralization of DENV2 and ZIKV in vitro by immune sera. Both naive and DENV2-immune mouse sera were inactivated for 30 min at 56 °C. In brief, sera were diluted in 96-well round bottom plates to 1:10, 1:30, 1:90, 1:270, 1:810, 1:2430,

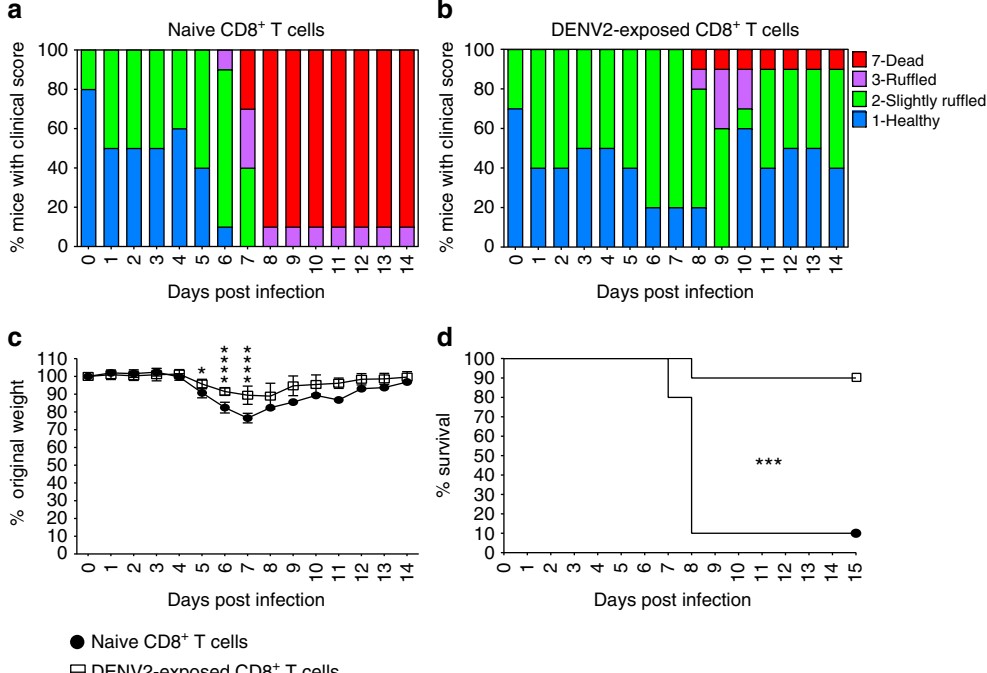

**Fig. 5** Decreased ZIKV-induced morbidity and mortality in *Ifnar1*⁻/⁻ mice adoptively transferred with DENV2-exposed CD8⁺ T cells. Naive or DENV2-exposed CD8⁺ T cells (1 × 10⁷) from *Ifnar1*⁻/⁻ mice were adoptively transferred via the retro-orbital route to recipient *Ifnar1*⁻/⁻ mice. One day later, mice were retro-orbitally challenged with 1 × 10³ FFU of ZIKV FSS13025. Clinical score (**a**, **b**), weight loss (**c**), and survival rates (**d**) were recorded daily. Mice having >20% weight loss and/or exhibiting paralysis of two hind limbs were euthanized according to Animal Protocols. This experiment was performed twice with $n = 5$ mice per group per experiment. Data represent pooled data from 10 mice per group and expressed as mean ± SEM. *$p < 0.05$, ***$p < 0.001$, ****$p < 0.0001$. **c** Two-way ANOVA. **d** Log rank test. Supplementary Tables 1 and 2 give exact values of $n$ and $p$

and 1:7290. DENV2 S221 or ZIKV FSS13025 was added to diluted sera (2 × 10⁴ FFU of DENV2/ZIKV per well). After 1 h incubation at 37 °C, U937 DC-SIGN cells were seeded in each well (1 × 10⁵ cells/well). The plates were incubated for 2 h at 37 °C (rocking every 15 min). Cells without serum were used as positive control. After incubation, the plates were centrifuged for 5 min at 300 × *g*, and supernatants were aspirated. Fresh medium was added to plates and incubated for 16 h at 37 °C. After fixation and permeabilization using Cytofix/Cytoperm solution (BD Biosciences), cells were stained with anti-CD209 PE (a dilution of 1:100) and a mixture of 4G2 FITC/2H2 FITC (1 μg/ml). Cells were analyzed on an LSRII flow cytometer (BD Biosciences, CA, USA) and the percentages of infected cells were determined using FlowJo software X 10.0.7. Percent inhibition for diluted serum was calculated using Eq. (1):

$$\text{Percent inhibition} = \frac{(\text{Positive control} - \text{test})}{\text{Positive control}} \quad (1)$$

**ZIKV challenge of mice transferred with DENV-immune serum.** Naive and DENV2-immune mice were sacrificed and sera were collected. Pooled naive and DENV2-immune *Ifnar1*⁻/⁻ mice sera were transferred passively (via retro-orbital route) to 5-week-old naive *Ifnar1*⁻/⁻ mice (600, 300 μl naive sera/mouse; 600, 300, 100, 33 μl DENV2-immune sera/mouse). Pooled naive and DENV2-immune sera from WT mice were retro-orbitally transferred to 2-week-old or 4-week-old naive WT mice (200, 75 μl naive sera/mouse; 200, 75, 25, 8.3 μl DENV2-immune sera/mouse) that had received 1 or 2 mg IFNAR1-blocking mAb MAR1-5A3 (intraperitoneal route) 1 h prior to serum transfer. One day after serum transfer, mice were infected retro-orbitally with 1 × 10⁴ FFU, 5 × 10⁴ FFU, or 1 × 10⁶ FFU of ZIKV FSS13025. Three days after challenge, mouse serum, spleen, liver, testis, brain, and eye were harvested.

**ZIKV challenge of mice transferred with DENV-exposed T cells.** Naive and DENV2-immune *Ifnar1*⁻/⁻ or WT mice were sacrificed and CD8⁺ T cells were isolated from splenocytes using magnetic positive selection Ab-coated microbeads (MiltenyiBiotec). Naive and DENV2-exposed CD8⁺ T cells were adoptively transferred to naive *Ifnar1*⁻/⁻ or WT mice (1 × 10⁷ naive CD8⁺ T cells, 1 × 10⁷ DENV2-exposed CD8⁺ T cells/mouse). One day later, *Ifnar1*⁻/⁻ mice were infected retro-orbitally with 1 × 10⁴ FFU of ZIKVFSS13025, whereas WT mice with IFNAR blockade (via intraperitoneal injection of 2 mg IFNAR1-blocking Ab MAR1-5A3 1 day before ZIKV challenge) were infected retro-orbitally with 1 × 10⁶ FFU of ZIKV FSS13025. Three days after ZIKV challenge, mouse serum, spleen, liver, testis, brain, and eye were harvested.

**DENV-exposed CD8⁺ T cell adoptive transfer and survival study.** Naive and DENV2-immune *Ifnar1*⁻/⁻ CD8⁺ T cells were injected retro-orbitally into naive *Ifnar1*⁻/⁻ recipients (1 × 10⁷ naive CD8⁺ T cells/mouse, 1 × 10⁷ DENV2-exposed CD8⁺ T cells/mouse). One day later, mice were inoculated retro-orbitally with 1 × 10³ FFU of ZIKV FSS13025. Weight loss and death (or moribund condition mandating euthanasia) were recorded on each day.

**Intracellular cytokine staining.** Splenocytes were plated (1 × 10⁶ per well, in 200 μl) in 96-well round-bottom plates and stimulated with 1 μg of individual peptide for 6 h in the presence of BFA and anti-CD107a PE (for the last 5 h, a dilution of 1:100). Cells stimulated with PMA/Ionomycin (P/I) (a dilution of 1:500) and cells without stimulation were used as positive and negative controls, respectively. After incubation, cells were stained with Abs against CD3 (a dilution of 1:100) and CD8 antigens (a dilution of 1:100). After fixation and permeabilization using Cytofix/Cytoperm buffer, cells were incubated with anti-IFNγ (a dilution of 1:200) and anti-TNF mAb (a dilution of 1:200). Samples were processed on a LSRII flow cytometer and analyzed using FlowJo software X 10.0.7.

**Viral burden analysis.** FFA was performed with BHK-21 cells as described[39]. Mouse organs (in 1 ml MEM medium) were homogenized for 3 min using Tissuelyser II (Qiagen Inc.) and then centrifuged for 5 min at 400 × *g*. 100 μl supernatant or serum or DENV2/ZIKV viral suspension was diluted serially and added to cells in duplicate and incubated for 1 h. After aspirating supernatant, wells were overlaid with 1% CMC medium. Two and half days after infection, cells were fixed with 4% formalin (Fisher Chemicals), permeabilized with 1% Triton X, and blocked with 10% FBS/PBS solution. Plates were incubated with the pan Flavivirus-specific mAb 4G2 (1 μg/ml), and then HPR-conjugated goat anti-mouse IgG (a dilution of 1:1000). Foci were developed using True Blue substrate and counted manually.

**Statistical analysis.** All data were analyzed using Prism 6 software (GraphPad Software, Inc.) and expressed as mean ± SEM. Grubbs' test was performed to detect outliers. A Kruskal–Wallis one-way ANOVA was used for multiple comparisons. Comparison between two groups was performed using Mann–Whitney test. Weight loss was analyzed by two-way ANOVA. Survival data were analyzed using the log rank test. $P < 0.05$ was deemed a statistically significant difference.

**Data availability.** The data that support the findings of this study are available from the corresponding author on request.

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

## Acknowledgements

This research was funded by NIAID/NIH grants 1R01 AI116813 (to S.S.), and R01 AI104972 and P01 AI106695 (to M.S.D.) and the La Jolla Institute for Allergy & Immunology institutional support to S.S. We would like to thank Anila Mamidi and Matthew Young for breeding mice and managing the survival study.

## Author contributions

J.W. and S.S. designed the project and experiments. J.W., A.E.N., J.A.R.-N., M.J.G. and K. K. performed the experiments. J.W., K.K. and S.S. wrote the manuscript. M.S.D. and S.S. edited the paper.

## Additional information

**Competing interests:** The authors declare that M.S.D. is a consultant for Inbios, Visterra, Aviana, and Takeda Pharmaceuticals and on the Scientific Advisory Boards of Moderna and OvaGene. The remaining authors declare no competing financial interests.

