## [Peer Review File · Nature Communications]

REVIEWERS' COMMENTS:

Reviewer #1 (Remarks to the Author):

The authors have revised the manuscript based on our comments and it is acceptable for publication in NCOM

Reviewer #2 (Remarks to the Author):

The author's examine the role of prior dengue immunity on subsequent ZIKV challenge using immunodeficient *Ifnar1*^{-/-} mice. While antibodies are generated in response to DENV immunization that cross-react with ZIKV, they are poorly neutralizing to ZIKV *in vitro* (1:15) and are not able to reduce titers significantly in ZIKV challenged mice. However, adoptive transfer of cross-reactive CD8 T cells is able to reduce ZIKV viral titers in challenged immunodeficient mice.

The author's state (lines 76-78) that the major questions that are unanswered to date regarding the role of prior immunity influencing ZIKV infection: Does previous DENV exposure confer cross-protection against ZIKV, as observed in the context of heterotypic reinfection with a different DENV serotype? If so, then what are the roles of cellular vs. humoral immunity in mediating this cross-protection against ZIKV?

In a recently published manuscript by the same PI, Wen et al (*Nat Microbiology* 2017), this question has already been answered for the most part. B07 transgenic *IFN*^{-/-} mice were immunized with DENV and challenged with ZIKV. "ZIKV/DENV cross-reactive CD8⁺ T cells in DENV-immune mice expanded post ZIKV challenge and dominated in the subsequent CD8⁺ T cell response. ZIKV challenge following immunization of mice with ZIKV-specific and ZIKV/DENV cross-reactive epitopes elicited CD8⁺ T cell responses that reduced infectious ZIKV levels, and CD8⁺ T cell depletions confirmed that CD8⁺ T cells mediated this protection". This was taken from the abstract of the *Nat. Microbiology* paper. So it is not clear what significant advances have been made from their previous manuscript.

The anti-DENV Ab response can have three different effects – protection, enhancement or no effect as stated by the authors. The major effect that determines the outcome of infection include epitope specificity, concentration of Ab and avidity. Sera that were used for the adoptive transfer studies were initially tested for binding to and neutralization of ZIKV *in vitro*. A major factor to be considered is the timing after DENV infection when people are exposed to ZIKV. A number of publications overlook this aspect and have added to the confusion in the literature regarding the role of prior T and B cell immunity with 1 flavivirus (dengue in this manuscript) impacting the outcome (positive, negative or no impact) of a more distantly related flavivirus such as ZIKV. In this study, adoptive transfer studies and challenge experiments were performed 1 month after DENV immunization and the timing represents 1 possible scenario of prior DENV immunity impacting ZIKV challenge using this immunodeficient mouse model. For a more complete picture, comparisons at 6 months and 1 year post DENV immunization will be more relevant and representative of immunity in people in DENV endemic areas who are exposed to ZIKV.

REVIEWERS' COMMENTS:

Reviewer #1 (Remarks to the Author):

The authors have revised the manuscript based on our comments and it is acceptable for publication in NCOM.

We thank the reviewer and greatly appreciate this positive comment.

Reviewer #2 (Remarks to the Author):

The author's examine the role of prior dengue immunity on subsequent ZIKV challenge using immunodeficient *Ifnar1*^{-/-} mice. While antibodies are generated in response to DENV immunization that cross-react with ZIKV, they are poorly neutralizing to ZIKV *in vitro* (1:15) and are not able to reduce titers significantly in ZIKV challenged mice. However, adoptive transfer of cross-reactive CD8⁺ T cells is able to reduce ZIKV viral titers in challenged immunodeficient mice.

The author's state (lines 76-78) that the major questions that are unanswered to date regarding the role of prior immunity influencing ZIKV infection: Does previous DENV exposure confer cross-protection against ZIKV, as observed in the context of heterotypic reinfection with a different DENV serotype? If so, then what are the roles of cellular vs. humoral immunity in mediating this cross-protection against ZIKV?

In a recently published manuscript by the same PI, Wen et al (*Nat Microbiology* 2017), this question has already been answered for the most part. B07 transgenic *IFN*^{-/-} mice were immunized with DENV and challenged with ZIKV. "ZIKV/DENV cross-reactive CD8⁺ T cells in DENV-immune mice expanded post ZIKV challenge and dominated in the subsequent CD8⁺ T cell response. ZIKV challenge following immunization of mice with ZIKV-specific and ZIKV/DENV cross-reactive epitopes elicited CD8⁺ T cell responses that reduced infectious ZIKV levels, and CD8⁺ T cell depletions confirmed that CD8⁺ T cells mediated this protection". This was taken from the abstract of the *Nat. Microbiology* paper. So it is not clear what significant advances have been made from their previous manuscript.

We appreciate the reviewer's comment. The goals of the published study with HLA transgenic mice were to identify human MHC class I-restricted epitopes in ZIKV and determine whether these epitope-specific CD8⁺ T cells could contribute to protection against ZIKV by immunizing mice with synthetic ZIKV epitope-specific and DENV cross-reactive peptides. In contrast, the major objective of the present study was to dissect the role of cellular vs. humoral immune response induced by prior DENV infection in protection against or pathogenesis of subsequent ZIKV infection using a sequential DENV-ZIKV infection model.

The anti-DENV Ab response can have three different effects – protection, enhancement or no effect as stated by the authors. The major effect that determines the outcome of infection include epitope specificity, concentration of Ab and avidity. Sera that were used for the adoptive transfer studies were initially tested for binding to and neutralization of ZIKV *in vitro*. A major factor to be considered is the timing after DENV infection when people are exposed to ZIKV. A number of publications overlook this aspect and have added to the confusion in the literature regarding the role of prior T and B cell immunity with 1 flavivirus (dengue in this manuscript) impacting the outcome (positive, negative or no impact) of a more distantly related flavivirus such as ZIKV. In this study, adoptive transfer studies and challenge experiments were performed 1 month after DENV immunization and the timing represents 1 possible scenario of prior DENV immunity impacting ZIKV challenge using this immunodeficient mouse model. For a more complete picture, comparisons at 6 months and 1 year post DENV immunization will be more relevant and representative of immunity in people in DENV endemic areas who are exposed to ZIKV.

We thank the reviewer for this comment and agree that future studies should be conducted to determine the duration of cross-protection. We have added the following sentence in the Discussion: Studies using mice that have been exposed to DENV for greater lengths of time than 1 month (such as 3, 6, and 12 months) may help answer questions related to the window of cross-protection.